# Effects of Motivational Climate on Fear of Failure and Anxiety in Teen Handball Players

**DOI:** 10.3390/ijerph17020592

**Published:** 2020-01-16

**Authors:** Manuel Gómez-López, Carla Chicau Borrego, Carlos Marques da Silva, Antonio Granero-Gallegos, Juan González-Hernández

**Affiliations:** 1Department of Physical Activity and Sport, Faculty of Sport Sciences, University of Murcia, Santiago de la Ribera, 30720 Murcia, Spain; mgomezlop@um.es; 2Campus of International Excellence “Mare Nostrum”, University of Murcia, 30720 Murcia, Spain; 3Sport Sciences School of Rio Maior, Polytechnic Institute of Santarém, Research Center in Life Quality (CIEQV), 2040-413 Rio Maior, Portugal; ccborrego@esdrm.ipsantarem.pt (C.C.B.); csilva@esdrm.ipsantarem.pt (C.M.d.S.); 4Department of Education, Faculty of Education Sciences, University of Almeria, 04120 Almeria, Spain; 5Health Research Center, University of Almeria, 04120 Almeria, Spain; 6Department of Personality, Evaluation and Psychological Treatment, University of Granada, 18071 Granada, Spain; jgonzalez@ugr.es

**Keywords:** coach, motivation, team sports, failure, well-being, performance

## Abstract

The aim of this study was to examine the effects of the motivational climate created by the coach and perceived by a group of young, high-performance handball players on their fear of failure and precompetitive anxiety. The study participants were 479 young handball players. The age range was 16–17 years old. Players were administered a battery composed of a Perceived Motivational Climate in Sport Questionnaire, a Performance Failure Appraisal Inventory, and Competitive State Anxiety Inventory-2R, to measure the aforementioned theoretical constructs. Using structural equation modelling (SEM), the results showed that the handball players experienced high levels task-involving climate and moderate values of self-confidence. In contrast, players experienced lower levels of ego-involving climate, fear of failure, and cognitive and somatic anxiety. The obtained model shows that fear of failure positively mediates the association between an ego-involving climate with both cognitive and somatic anxiety, and negatively in terms of self-confidence. In contrast, fear of failure does not mediate the associations between a task-involving climate and both somatic and cognitive anxiety and self-confidence.

## 1. Introduction

In sport, physical and psychological factors that affect the athlete’s performance. While motivational factors play an important protector role in the long-term effects of regular sport practice on the psychosocial development of the athlete, the emotional responses provoked by anxiety can affect athletic performance [1], and a state of permanent anxiety can even become a determining factor in the relative or total abandonment of sports practice [2]. Anxiety is a psychological response produced as a consequence of differences between the self-assessment of a person’s ability to respond to a specific situation (e.g., challenge, action) and that person’s actual ability to respond [3]. Understood as a neurotic personality feature [4], anxiety is associated with the tendency to show poor emotional adjustment in the form of high levels of reactivity and emotional sensitivity (e.g., fear, worry or anger) [5]. Despite this, anxiety also exerts a self-regulatory influence against low levels of coping or motivation [6], which can set up adaptive effects which are suitable for short-term performance improvements [7].

In a context of skill and achievement, in which athletes try to reach a goal and in which proving competence and ability is important [8], Roberts and Treasure [9] claim that physical and psychological well-being depends on the contexts where sports practice is carried out. More specifically, in the case of a team sport, such as handball, and at the height of adolescence, peers and coaches are the main social agents in a team [10,11] in terms of the motivational climate perceived during training [12,13,14,15,16]. As result of that self-evaluation cognitive process, and mediated for other psychological skills (e.g., perceived control or coping resources), it is not the perceived situations that directly cause the anxiety response, but certain situations of pressure or potential conflict that become stressful when they are assessed as threatening [17].

According to achievement goal theory [18,19], this motivational climate refers to the set of signals generated by family, friends or colleagues, a coach, etc., which are perceived by athletes in their environment, and through which the key factors for success and failure are identified. That is, it depends on how the context is subjectively interpreted in terms of the present criteria of success or failure. Thus, according to achievement goal theory [19], either an ego-involving or a task-involving motivational climate will be created according to how the athletes perceive the context [20,21]. In this regard, if victory and showing good skills and performance are the most important aspects for the people in the players’ environment, they will create an ego-involving climate, while if they see effort, personal improvement and skills development as central aspects, they will create a task-involving climate. One of the triggers of avoidance behaviors and fear of failure is the motivational climate created during adolescents’ socialization processes. An athlete’s ability to manage and control this situation depends on many factors, and poor management of it can lead to errors or mistakes [22].

Even when sports are one of the most suitable contexts for achievement, as athletes are motivated to succeed showing their competence and skills [19,23], different authors claim that it is also a medium that can reflect athletes’ incompetence in front of their peers [24]. This is related to the fact that many athletes experience fear of failure in highly competitive environments (e.g. national or regional tournaments), when their performance is judged by others (e.g., referees, coaches, parents, peers, or spectators) on the basis of success criteria [25]. This motivational climate of ego involvement emphasizes public recognition based on social comparison through success, and considers error as something to be avoided [3]. Thus, the practice of sports can generate a sensation of fear of failure and feelings of shame, which, in turn, create insecurity, anxiety, and stress in athletes, as well as avoidance behaviors in relation to what people may say or think about them, especially during the course of a game, which results in impaired performance [26]. 

Fear of failure is conceptualized as the motive to avoid failure in evaluative achievement situations, associated with anticipatory shame [27]. Conroy, Poczwardowski, and Henschen [28], and Conroy, Willow and Mezler [29] define fear of failure as a stable tendency to anticipate shame and humiliation after failure. In the context of sport, Conroy [30] and Conroy et al. [29] developed a multidimensional, hierarchical model of fear of failure based upon the cognitive-motivational-relational theory of emotion [31]. 

Different studies have shown that there is a positive relationship between the perception of a coach-initiated ego involvement motivational climate and the fear of failure [15,16,22,32,33]. Fear of failure triggers an assessment of the negative consequences (e.g., ridicule, shame, or stage fright) on the player’s well-being of those potential emotions that players may feel after not performing a task correctly in front of their peers, coaches, or teachers [25,29,30]. Gustafsson, Sagar, and Stenling [34] recently showed that the fear of feeling shame is associated with high levels of psychological stress, which would have an impact on fear of failure. Youngsters with a high level of competitive anxiety worry more than their peers who have low levels of anxiety about making mistakes, not playing well, or losing. Within the context of sport, athletes with higher anxiety are also more concerned about the evaluation of their coaches, peers, and parents, and have strong expectations that failure will lead to criticism from them [35,36,37,38]. Some elite athletes may be reluctant to reveal their fear of failure as they associate that fear with weakness of character and lack of success and confidence [25]. Fear of failure, therefore, affects the well-being, interpersonal behavior and sporting performance of athletes [39]. 

Because the coach is the highest authority on the team, his/her decisions and an agonizing (or highly competitive) motivational climate are more likely to induce anxiety in young participants, probably experienced when the focus (personal or situational) is on the player’s skill level and public recognition of his/her achievements [3,19], and his/her emotional and cognitive self-assessments are focused on the potential punishment for making mistakes [40]. The literature has shown that athletes’ perceptions of an ego-centered climate are positively correlated with competitive anxiety [41,42]. According to West, Rhoden, et al. [43], neuroticism has a strong positive relationship with amotivation, and it also deepens in the influence of the extroversion factor in athletes’ motivational processes. Athletes who perceive a high level ego-centered motivational climate had the highest levels of cognitive anxiety, especially in the form of worry [44,45,46]. While somatic anxiety can affect the motor component of sports behavior, cognitive anxiety can have a greater impact on athletes’ attention spans, concentration, and decision making abilities [47]. This cognitive component has two levels: the first refers to concerns about the potentially negative consequences associated with performance; and the second, deconcentration, describes difficulties in focusing on the key aspects of the competitive task [48]. Finally, in order to stand out, these sportsmen tend to be more prone to responding to errors and unsatisfactory performance with punitive responses [49].

Another important aspect is that players who perceive a coach task-involving climate in their training see defeat as a learning factor, and believe that success is achieved with effort, as opposed to those who were subject to an ego climate, for whom ability and tricks are the main tools by which to succeed in sports [1,21]. Furthermore, coaches who promote a task-involving climate see mistakes as part of players’ learning processes, a stance that is transmitted to their players, who thus feel lower levels of fear of failure, as recently shown by Gómez-López, Ruiz-Sánchez, and Granero-Gallegos [15] in a study with handball players. On the other hand, the literature has shown that athletes’ perceptions of a master’s climate are associated with low levels of anxiety [41,46,50,51].

In the present study, the motivational climate generated by the coach was selected, as he/she represents the hierarchical figure of reference in the sporting context [52]. In addition, as the leader of the sports group, the coach plays a prominent role in the process of sport socialization due to his/her ability to judge and provide rewards or punishment to athletes [53]. This is very evident in athletes aged 16 to 17 years, who see the coach as a model that helps them to develop their autonomy, competence, commitment, and relationship through their behavior and instructions. [2,54]. The selection of the sample age study was motivated by the crucial role played by the coach at this stage of development.

The aim of this study is to analyze how a motivational climate generated by the coach affects the fear of failure and the precompetitive anxiety perceived by handball players in regional teams. Specifically, we hypothesize that players’ perceptions of mastery-involving climate will be positively related to self-confidence, while perceptions of an ego-involving climate will be positively related to fear of failure and precompetitive anxiety. 

Therefore, the present study, by examining the relationship between perceived motivational climate, fear of failure, and anxiety in sport, will broaden the knowledge base on motivational climate and fear of failure as a background to anxiety in competitive situations, and will empirically recognize the coach-initiated motivational climate and fear of failure as a predictor of anxiety in sport.

## 2. Materials and Methods

### 2.1. Participants 

Four hundred and seventy-nine youth handball players (250 boys and 229 girls), who were selected to compete in the Spanish Regional Championships, participated in the study. Since they are the best players of their regions, they belong to the regional handball team, and most of them have participated in more than two Spanish handball championships. These players were rated as “high-performance players” by the Spanish Sports Council according to Royal Decree 971/2007 (July 13, 2007). The age range was 16 (40.1%) and 17 (59.9%) years old (M = 16.60; SD = 0.50.) With regard to the years of experience, 85.4% had more than five years of experience as federated handball players.

### 2.2. Measurement instruments

Perceived Motivational Climate in Sport Questionnaire (PMCSQ-2) [20,51]. The Spanish version of PMCSQ-2 was used [10,11]. The inventory includes 29 items grouped into two dimensions measuring the ego-involving (competitive) climate (14 items, e.g. “On this team, the coach gives most of his or her attention to the stars.”), and the task-involving (mastery) climate (15 items, e.g. “On this team, the coach emphasizes always trying to do your best”). Each item was headed with the phrase “In my training group or team…”. Answers were collected on a Likert-type scale ranging from “strongly disagree” (1) to “strongly agree” (5). Cronbach´s alpha (α) for the instruments range from α = 0.82 in mastery or task-involving climate, to α = 0.85 in competitive or ego- involving climate.

Performance Failure Appraisal Inventory (PFAI) [29]. The Spanish long version of the PFAI [22] includes 25 items grouped into five dimensions: Fear of Experiencing Shame and Embarrassment (e.g. “When I am failing, it is embarrassing if others are there to see it.”), Fear of Devaluing One’s Self-estimate” (e.g. “When I am failing, it is often because I am not smart enough to perform successfully”), Fear of Having an Uncertain Future (e.g. “When I am failing, I believe that my plans for the future will change”), Fear of Important Others Losing Interest (e.g., “When I am not succeeding, some people are not interested in me anymore.”), and Fear of Upsetting Important Others (e.g. “When I am failing, important others are disappointed.”). All items were headed by the phrase “In my sports practice…”. The answers were collected on a Likert-type scale ranging from “do not believe at all” (1) to “believe 100% of the time” (5). Here, the internal consistency analysis was satisfactory for the different subscales; Fear of Experiencing Shame and Embarrassment, α = 0.85; Fear of Devaluing One’s Self-Esteem α = 0.70; Fear of Having an Uncertain Future, α = 0.83; Fear of Important Others Losing Interest, α = 0.84; Fear of Upsetting Important Others, α = 0.81. Fear of failure, α = 0.74.

Competitive State Anxiety Inventory-2R (CSAI-2R) [55]. This inventory comes from the original CSAI-2 composed of 27 items [56]. This study used the Spanish version of the revised edition which was validated on Spanish athletes [57]. The CSAI-2R is a specific inventory for the sports context, but with a one-dimensional conception of the state of anxiety. The scale is composed of 16 items grouped into three dimensions that measure somatic state anxiety (6 items; e.g., "My heart is racing"), self-confidence (5 items; e.g., "I’m sure of myself"), and cognitive state anxiety (5 items; e.g., "I’m worried that others will be disappointed with my performance"). Each item on the scale had to be suitable to the measure of the player’s feelings just before the competition. The answers were collected on a Likert scale of 4 points ranging from “not at all” (1) to “very much so” (4). The internal consistency analysis was satisfactory for the different subscales; somatic state anxiety, α = 0.78; self-confidence, α = 0.82; cognitive state anxiety, α = 0.75.

### 2.3. Procedure

The study was carried out during the Championship of Autonomic Selections. The national federation, the regional federations, and the coaches of the different regional teams all granted permission prior to our data collecting after reading a letter explaining the objectives of the study and the way it would be carried out. Prior to the administration of the questionnaires to the participants, and in accordance with the ethical guidelines of the American Psychological Association (APA), they were presented with an informed consent form [15] for ethical compliance and data protection, ensuring the rigor of the investigation and the privacy of the information obtained. The consent obtained from the players and their parents or tutors was both written and informed. A sample of the questionnaire was provided to each participant. Data collection was carried out during the national Championship, at the hotels where the teams were staying during time off, in agreement with the coaches and in the presence of one of the researchers. Each participant had 20–30 minutes to complete the questionnaire, and they were all briefed on the objectives of the study, their rights as participants, the voluntary nature of their participation, and the confidentiality of answers and data management. They were also informed that there were no correct or incorrect answers, and were asked to give true and honest replies. Following data verification, the following variables were recorded: gender, year of birth, years of experience as a handball player, playing position, and the numbers of hours per week dedicated to training, as well as the times it was carried out. The protocol was approved by the Ethics Committee of the Universidad de Murcia (ID: 1494/2017). All subjects gave written informed consent in accordance with the Declaration of Helsinki [58].

### 2.4. Statistical Analysis

All variables were analyzed for means, standard deviation, and bivariate correlations, and, as suggested by Kline [59], a two-step maximum likelihood (ML) approach was performed in AMOS 23.0 (SPSS Inc., Chicago, IL, USA) In order to analyse the psychometric properties of the purposed model, a confirmatory factor analysis (CFA) was performed. Internal consistency was accessed using Raykov [60] composite reliability with a cut-off value of 0.70 [61,62]. Convergent validity was estimated considering average variance extracted (AVE) higher than 0.5 [61]. Discriminant validity was established when the correlation coefficients were lower than the AVE for each construct exceeding the squared correlations between that construct and any other [63]. A structural equation modelling (SEM) was also performed in order to test the relations between different constructs. The following indices were specifically used for the CFA and SEM analyses: Comparative Fit Index (CFI), Tucker Lewis Index (TLI), Standard Root Mean Residual (SRMR), and Root Mean Square Error of Approximation (RMSEA) with a Confidence Interval (CI: 90%). For these indices, scores of CFI and TLI ≥0.90, SRMR and RMSEA ≤0.8 were considered as acceptable, following several recommendations [61,64,65].

### 2.5. Mediation analysis

For mediation analysis, the direct and indirect effects among constructs on outcome variables were analyzed as suggested by Hair et al. [61] and Hayes [66]. Bootstrap resampling was performed (1000 samples) via AMOS 23.0 A 95% confidence intervals (CI) was used to analyse the significance of direct and indirect effects. The indirect effect was considered significant if the CI did not include/cross zero, [66,67,68]. The effect sizes were evaluated as trivial (0–0.19), small (0.20–0.49), medium (0.50–0.79), and large (0.80 and greater), as suggested by Cohen [69].

## 3. Results

### 3.1. Preliminary analyses

A preliminary analysis presented no missing values or outliers (univariate or multivariate). The data also revealed no univariate distributions violations. However, Mardia´s coefficient form multivariate kurtosis exceeded the recommended value (>0.5). Thus, Bootstrap Bollen-Stine [70] was performed [71]. Additionally, the variance inflation factor (VIF) was calculated to check the collinearity diagnosis. Results indicated that all VIF values were less than 10, i.e., a recommended value suggested by Hair et al. [62]. Finally, a G*Power 3.1 [72], was used to determine the required sample size, considering the following parameters: effect size f_2_ = 0.1; a = 0.05; statistical power = 0.95 and four predictors, given that the minimum required size would be 176 subjects, which was respected in the present study.

### 3.2. Measurement model

Table 1 shown the means, standard deviations, and bivariate correlations among all of the studied variables. The players demonstrated high levels of task-involving climate (*M* = 3.17; *SD* = 0.56), and moderate values of self-confidence (*M* = 2.77; *SD* = 0.63). In contrast, players showed lower levels of ego-involving climate (*M* = 1.63; *SD* = 0.76), fear of failure (*M* = 1.94; *SD* = 0.73), cognitive (*M* = 2.02; *SD* = 0.49), and somatic anxiety (*M* = 1.58; *SD* = 0.41). The correlation patterns evidence that a task-involving climate is negatively and significantly associated with an ego-involving climate and fear of failure, while an ego-involving climate is positively and significantly associated with all variables under analysis. In turn, fear of failure was positively and significantly associated both cognitive and somatic anxiety and negatively associated with self-confidence. Regarding anxiety constructs, the results showed that somatic and cognitive anxiety were associated positively and significantly, while self-confidence was negatively and significantly associated with both somatic and cognitive anxiety. Finally, it was possible to observe that all constructs present an adjusted value of composite reliability, with all of them being greater than or equal to 0.70 [61]. The test of the measurement model included the task and ego-involving climate, fear of failure, cognitive, somatic, and self-confidence. The results showed that the initial model did not fit the data (χ2 = 1105.05 (545); SRMR = 0.059; B-Sp ≤0.001; RMSEA = 0.046 (90%CI = 0.043, 0.050); TLI = 0.881; CFI = 0.891). Furthermore, modification indexes from AMOS 23.0 were analyzed, revealingthat errors from items 11 and 15 (ego-involving climate) and items 6 and 12 (somatic anxiety) should be correlated. Theoretically, these relationships make sense due to the multidimensional association of these indicators with the latent variable, maintaining the parsimoniosity of the model. In these circumstances, an alternative model should be considered, provided that its theoretical integrity, parsimony, and adjustment to data is maintained [59]. After error correction, the respecified model fit the data (χ2 = 988.818 (543); SRMR = 0.057; B-Sp ≤ 0.001; RMSEA = 0.042 (90%CI = 0.037, 0.046); TLI = 0.905; CFI = 0.913). Additionally, the measurement model revealed that no problems occurred in terms of the convergent and discriminant validity, since the average extracted variance was greater than or equal to 0.50 [61,63], and the square correlations among all constructs were less than the AVE of each factor [63].

### 3.3. Structural model

The structural model showed a good fit with the data (χ2 = 1032.261 (295); SRMR = 0.064; B-Sp ≤0.001; RMSEA = 0.043 (90%CI = 0.039, 0.047); TLI = 0.900; CFI = 0.906). For the standardized direct effect (Figure 1), positive and significant effects were observed across ego-involving climate and fear of failure (*β* = 0.37 (0.226, 0.489)) and fear of failure, i.e., both somatic (*β* = 0.44 (0.306, 0.557)) and cognitive anxiety (*β* = 0.51 (0.375, 0.621)). In contrast, a negative but nonsignificant effect was observed between a task-involving climate and fear of failure (*β* = −0.09 (−0.231, 0.071)) and between fear of failure and self-confidence (*β* = −0.22 (−0.362, −0.088)).

Regarding mediation analysis between task and an ego-involving climate on cognitive, somatic, and self-confidence, the results showed a positive a significant indirect effect across ego-involving climates, i.e., in terms of both somatic (*β* = 0.16 (0.087, 0.273)) and cognitive anxiety (*β* = 0.19 (0.113, 0.237)), through fear of failure, and a negative and significant effect between ego-involving climate and self-confidence (*β*= −0.08 (−0.140, −0.035)), via fear of failure. In turn, the task-involving climate evidenced a negative but nonsignificant indirect effect with somatic (*β* = −0.04 (−0.103, 0.028)) and cognitive anxiety (*β* = −0.05 (−0.118, 0.031)) through fear of failure, and a positive but nonsignificant indirect correlation with self-confidence (*β* = 0.02 (−0.012, 0.064)), also via fear of failure.

In summary, the mediation analysis revealed that fear of failure positively influences the association between an ego-involving climate, i.e., both cognitive and somatic anxiety, and negatively influences self-confidence. In contrast, fear of failure does not influence the associations between a task-involving climate and both somatic and cognitive anxiety and self-confidence.

## 4. Discussion

In a highly competitive sports environment, such as teams selections, many players fear failure due to the evaluation they are regularly subjected to, according to the criteria of performance and success, both by their coaches, parents, teammates, and even by fans that attend the competitions [12,15,16,25]. Smith, Smoll, and Schutz [47] suggested that the most salient sources of situational stress in competitive sports environments are related to performance failure. Increasing pressure to achieve top performances will unequivocally create anxiety and fear of failure in young athletes [26].

The results recently provided by Gómez-López et al. [15] showed that except for fear of feeling shame, which was predicted by the peer ego-involving climate, all the aversive causes of fear of failure may be predicted by the coach climate. The immediate environment was proven to be determinant in avoidance behaviors and fear of failure. For that reason, the aim of this study was to analyze how the motivational climate generated by the coach affected the fear of failure and precompetitive anxiety perceived by the handball players of the region selection’s teams.

The descriptive results showed that most of the handball players analyzed perceived a high motivational climate of task involvement as well as a moderate level of self-confidence. Similarly, there were also players who perceived lower levels of motivational climate of ego involvement, fear of failure, and anxiety, i.e., both somatic and cognitive anxiety. These are positive results in view of the fact that so far, the literature has shown that coaches who create a task-involving climate promote effort, interest in learning, and personal progress through comparison to oneself to value competence level, skill development, and mutual cooperation in their team players [73]. Furthermore, different studies have shown that sports players subject to this type of motivational climate enjoy psychological well-being, increase their enjoyment of practicing their sport and of their team’s performance, and show lower levels of competitive anxiety, as opposed to players training in a competitive or ego-based climate, who experience high anxiety levels and derive less satisfaction in sports practice [74].

The results of the correlation analysis are consistent with the hypothesis made, given that the perceived motivational climate is strongly correlated with the fear of failure; this is strongly correlated with sports anxiety. The mediation analysis revealed that fear of failure positively influenced the association between an ego-involving climate, both in terms of cognitive and somatic anxiety, and negatively in terms of self-confidence. In contrast, fear of failure did not have an influence upon the associations between a task-involving climate and both somatic and cognitive anxiety and self-confidence. These results showed that the hypothesis formulated in the study were not completely fulfilled, since the fear of failure did not influence the association between the motivational climate of task involvement and self-confidence. The results found by Tsai and Chen [33], regarding the nonsignificant results found between the fear of failure and the climate of involvement in the task, were similar.

Even so, these results are congruent with previously-published data. The coaches who promote a task climate see mistakes as part of players’ learning processes, a stance that is transmitted to their players who thus feel lower levels of fear of failure, as recently shown by Tsai and Chen [33] with Taiwanese athletes, and by Ruiz-Sánchez et al. [16,32] with Spanish handball players. Likewise, Moreno-Murcia, Huéscar Hernández et al. [75] showed a significant and positive relationship between coaches’ controlling style and athletes’ fear of failure, whereas coach autonomy support was associated with a reduced fear of failure.

On the other hand, the literature has shown that the players who trained in a competitive climate (ego climate) had higher levels of anxiety [46,47], lower satisfaction levels with sports practice [2,54], and a reduced intention to continue doing sport [14]. Previously, Ntoumanis and Biddle [45] found no significant direct link between motivational climate and competitive anxiety, thus implying that motivational climates could have an indirect impact on affective responses through the different goal orientations.

Previous research has shown that fear of failure is related to high levels of fatigue, psychological stress, concern, and sports anxiety [29,34]. Even more, it was recently demonstrated that it is a strong predictor of the sport anxiety [26]. Hence, it has been confirmed that fear of failure affects well-being, interpersonal behavior, and the performance of the players [39].

Therefore, the present study demonstrates that the coach’s role as a socializing agent is important regarding avoidance behaviors, fear of failure, and the perceived anxiety for the players during sports practice. According to the results of the present study, coaches should provide their players with a motivational climate of task involvement which valorizes success and failure as a means of learning, and not merely as a sporting outcome. Also, during the competition, the coach should emphasize achievements related to skill, personal effort, and interest in the sports activity itself, and not only in the results. In this way, players will enjoy the practice of sport, will reduce their fear of failure, and will increase their level of commitment.

As a limitation of the study, it should be mentioned that although the results extend existing, published information, the specificity of the sample examined limits its generalization. These results require confirmation in future research with individual athletes where the interaction with the coach is different. In addition, other variables that influence the anxiety or fear process from personal perception (e.g. impulsivity or neuroticism) or from social perception (e.g. parental or peer influence) were not taken into account, which might explain why the reflected relationships, although interesting, are only partially explained. Perhaps the use of data from only one sport (handball) may suggest that they are taken into account in studies with team sports, and may be contrasted with other studies involving sports carried out by individuals. 

## 5. Conclusions

It was shown that players’ immediate sports environment, created based on the coach’s motivational climate, is a critical factor in terms of fear of failure and the perception of anxiety in handball practice. The results obtained reinforce the importance of supportive behavior and promoting a coach task climate, both during training and in competition, as opposed to just results. Likewise, players should be involved in task-oriented criteria in order to analyze failures and successes, not only as a means to improve their results, but also to make them value errors as a means of learning, thus reducing their fear of failure and favoring satisfactory experiences and sports commitment. In conclusion, the present study is unique in determining the relationship between motivational climate, fear of failure, and sport anxiety in a sample of young male players from a team sport. 

Finally, and based on all that has been pointed out so far and on the extensive review carried out, it is necessary to highlight the importance of this present study since, in the first place, there are, to date, no published studies on sports in Spain attempting to analyze the three variables studied here. Further to this, it constitutes a preliminary study with which to guide future research in which the main objective would be to carry out individualized psychological interventions on coaches to improve satisfaction levels among their players. Further topics that could be addressed in the future are the influence on players of parents and peers, and the difference between an athlete’s and coach’s own perception of the coach-created motivational climate.

Research on the influence of personality features (e.g., neuroticism) on the psychological skills (e.g., self-control, self-talk) is increasingly necessary. For this reason, the results offer important data for the sport sciences, but they should be reinforced by studies that deepen the mediation and moderation of personality traits (e.g., high-order traits) and their relationship with functional (e.g., motivation or self-efficacy) and dysfunctional (e.g., anxiety or concerns) psychological responses.

We hope that the current findings will increase the general understanding of the important role of the motivational climate that the coach favors over the fear of failure and anxiety in sports environments.

## Figures and Tables

**Figure 1 ijerph-17-00592-f001:**
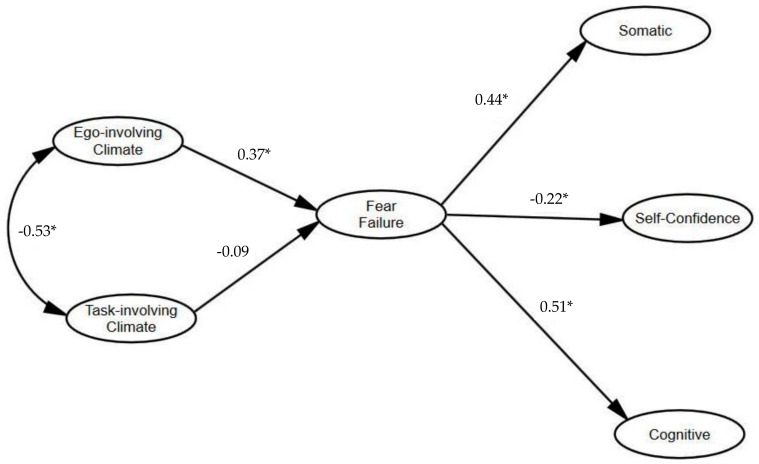
Standardized individual parameters – Hypothesized model. Note. (*) significant paths.

**Table 1 ijerph-17-00592-t001:** Descriptive and correlation analysis for all constructs and composite reliability.

Constructs	TI	EI	FF	Somatic	Self-Confidence	Cognitive
Task-involving	1	-	-	-	-	-
Ego-involving	−0.53 **	1	-	-	-	-
FF	−0.31 **	0.43 **	1	-	-	-
Somatic	−0.07	0.14 **	0.44 **	1	-	-
Self-Confidence	0.01	0.16 **	0.50 **	−0.20 **	1	-
Cognitive	−0.01	−0.19 **	−0.19 **	0.61 **	−0.38 **	1
Mean	3.17	1.63	1.94	1.58	2.77	2.02
SD	0.56	0.56	0.73	0.41	0.63	0.49
CR	0.82	JEL M ISLIS0.86	0.75	0.76	0.82	0.75

Note: TI = task-involving climate; EI = ego-involving climate; FF = fear of failure; CR = Composite reliability; SD = standard deviation; ** *p* ≤ 0.01.

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
