# Peer review of "Effects of Motivational Climate on Fear of Failure and Anxiety in Teen Handball Players"

_ijerph, 2020, doi:10.3390/ijerph17020592_

Round 1

Reviewer 1 Report

The study is quite interesting and provides new and therefore valuable information for your area of knowledge.

The manuscript is of sufficient quality to be published in this journal, although I would like to recommend the improvement of some aspects:

Attached are some suggestions included in the attached document. Include the practical proposals/applications of the results of the study, after the discussion and before the limitations section. Include also the "strengths" of the study and its results. What do you bring to the field of knowledge and other similar studies? Include future perspectives, or possible future studies, considering the limitations.

Author Response

Thank you very much for the suggestions that have improved the manuscript submitted. The practical applications of the results and limitations of the study have been added and highlighted in the last two paragraphs of the discussion. The importance and strengths of the results obtained have also been highlighted in the conclusions section, mentioning that this is the only study in Spain that analyses these variables in a sport. It is also highlighted what it is contributed to the field of knowledge of this specialty and the possible future research proposals that follow this study.

Finally, it should be noted that all the suggestions made by the reviewer in the attached manuscript have been accepted.

Reviewer 2 Report

The abstract is quite good, it is short and focusses on the main results.

Introduction

Line 4 -  delete "and"   (...and even a state…..)

Aim of study - Specify better the hypothesis of study (Specifically, we hypothesize that players perceptions mastery -involving climate will be positively related to self-confidence and that the perceptions of an ego-involving climate will be positively related to fear of failure and pre-competitive anxiety.) … Why??

Furthermore, the authors need to discuss power issues and comment on "why" handball (convenience sampling?). It is important to ensure confidentiality, and thus remove any identifiable and personal information from the manuscript.

Sample: Some information for the sample description is missing in the script so far. Probably the authors have assessed the information but simply forgot to mention them: training years (mean, SD, range) and training days per week. A time-line, based on the training plan, may illustrate a 'typical training week from Monday to Sunday' with all training sessions, including the time, and competitive match.

The Discussion is consistent. The conclusions are consistent with the evidence and arguments presented.

Author Response

Thank you very much for the suggestions that have improved the manuscript submitted. All suggestions made by the reviewer have been accepted. The objective and hypothesis have been improved and the confidentiality of the sample has been assured.

Regarding the selection of the sample, this sample was chosen because in Spain there are no studies of these characteristics with players belonging to team sports teams, and because some of the authors are specialists and work professionally in this sport.

The wording of the sample characteristics has been improved. It should be noted that we do not have any information regarding the number of training sessions or hours per session. Probably, these players, being the best of their region, train at least 2-3 times a week with their clubs and then during the weekend, they would have the match corresponding to their regional league together with some training with the regional team. We do not have this information, because, in addition, each regional Federation organizes as it believes convenient its planning of the Championship of Spain of Selections. There are Federations that concentrate on the pieces of training at weekends and others do it on the holidays of the year.

Reviewer 3 Report

Abstract:

Very well written, I have no comments.

Introduction:

It seems that there is a missing analysis of common variables that are pertinent.

State-trait anxiety should be discussed since it has quite a bit to do with motivational climate perception.

O’Rourke, D. J., Smith, R. E., Smoll, F. L., & Cumming, S. P. (2011). Trait anxiety in young athletes as a function of parental pressure and motivational climate: is parental pressure always harmful?. Journal of Applied Sport Psychology23(4), 398-412.

Also, trait neuroticism should be explored:

West, J., Rhoden, C., Robinson, P. D., Castle, P., & St Clair Gibson, A. (2016). How motivated are you? Exploring the differences between motivational profiles and personality traits. Sport and Exercise Psychology Review12(1), 28-42.

In the 5th paragraph third line, change “ridiculous” to ridicule, I believe that is what the authors mean.

In the 6th paragraph change “his” to their since coaches aren’t just males.

Materials and Methods:

The only variable that I wish the authors obtained would be a metric of physical fitness, because there is some theoretical reason to think that more fit individuals may perceive anxiety differently. Perhaps this could be discussed in the limitations. Also, what was the gender distribution of the handball players?

Results and Discussion:

I believe the authors did a good job here, just need to include a limitations section since some of the variables gathered may have overlap with better established psychological constructs such as Neuroticism and State/Trait Anxiety.

Author Response

Thank you very much for the suggestions that have improved the manuscript submitted. All suggestions made by the reviewer have been accepted.

The introduction and discussion with the reviewer's suggestions have been improved, including new references suggested by the reviewer. The wording of the characteristics of the sample has been improved. It should be pointed out that the physical fitness variable was not studied, so it has been added as a limitation of the study. The wording of the limitations of the study has been improved (end of the section on the discussion of the results). The wording of the conclusions has been improved.